# Seasonal Variations in Production Performance, Health Status, and Gut Microbiota of Meat Rabbit Reared in Semi-Confined Conditions

**DOI:** 10.3390/ani14010113

**Published:** 2023-12-28

**Authors:** Dingcheng Ye, Xiaoning Ding, Shuo Pang, Yating Gan, Zhechen Li, Qianfu Gan, Shaoming Fang

**Affiliations:** 1Institute of Animal Husbandry and Veterinary Medicine, Fujian Key Laboratory of Animal Genetics and Breeding, Fujian Academy of Agricultural Sciences, Fuzhou 350013, China; 13960756233@126.com; 2College of Animal Science (College of Bee Science), Fujian Agriculture and Forestry University, Fuzhou 350002, China; ding14784384628@outlook.com (X.D.); pangshuo9527@outlook.com (S.P.); gan0929@outlook.com (Y.G.); lizhechen1109@outlook.com (Z.L.)

**Keywords:** seasonal variations, meat rabbit, production performance, gut microbiota, health status, semi-confined conditions

## Abstract

**Simple Summary:**

Due to the fact that rabbit meat possesses high levels of proteins and polyunsaturated fatty acids with low contents of fat and cholesterol that can meet the urgent demand for a healthy diet in modern life, the meat rabbit industry in China has been rapidly developed in recent years. However, rabbits are more sensitive to seasonal changes owing to their specific physiological characteristics, such as few sweat glands, thick coats of fur, high metabolic rate, and high growth rate. In this study, we investigated the variations in productivity, health status, and gut microbiota of meat rabbits reared in semi-confined conditions between summer and winter. In the summer season, the relative high ambient temperature and humidity could trigger disturbance of the gut microbiome, potential heat stress, reduced antioxidant defense, and increased inflammation risk, which consequently deteriorated production performance. In the winter season, changes in energy demand, photoperiod, and feeding pattern should be regarded as important factors that affect the productivity of meat rabbits. Based on our findings, we not only proposed several realistic strategies to alleviate the unfavorable influences of seasonal alterations on the productivity and well-being of meat rabbits but also pointed out the future directions for this study of molecular mechanisms in adaptation physiology.

**Abstract:**

In this study, we investigated the variations in production performance, health status, and gut microbiota of meat rabbits raised in the semi-confined barn during summer and winter. Compared to summer, rabbits reared in winter possessed significantly higher slaughter weight and carcass weight. Rabbits fed in the summer were more vulnerable to different stressors, which led to increased protein levels of HSP90, IL-1α, IL-1β, IL-2, and concentrations of MDA, but declined GSH and SOD activities. Additionally, significant differences in gut microbial communities were observed. Compared to the winter, rabbits fed in the summer had significantly lower and higher alpha and beta diversity. Both Firmicutes and Verrucomicrobiota were the dominant phyla, and they accounted for greater proportions in the winter than in the summer. At lower microbial taxa levels, several seasonal differentially enriched microbes were identified, such as *Akkermansia muciniphila*, the *Oscillospiraceae NK4A214 group*, the *Christensenellaceae R-7 group*, *Alistipes*, and Muribaculaceae. Functional capacities linked to microbial proliferation, nutrient metabolism, and environmental adaptive responses exhibited significantly different abundances between summer and winter. Moreover, strong interactions among different indicators were presented. Based on our findings, we not only proposed several potential strategies to ameliorate the undesirable effects of seasonal changes on the productivity and health of meat rabbits but also underscored the directions for future mechanistic studies of adaptation physiology.

## 1. Introduction

Meat Rabbits are reared primarily for their meat, which is characterized by high levels of proteins and polyunsaturated fatty acids with low contents of fat and cholesterol [1]. Due to these specific nutritional characteristics, which can meet the urgent demand for a healthy diet in modern life, the meat rabbit industry in China has been rapidly developed in recent years. However, farm animals, including rabbits, commonly encounter a variety of biotic and abiotic stressors in their lifetimes, such as extreme temperatures, poor nutrition, overproduction, transportation, weaning, infection, and diseases [2]. These stressors adversely affect the productivity, well-being, and welfare of farm animals, which further leads to significant economic losses in animal husbandry. Compared to other farm animals, rabbits are more sensitive to seasonal changes owing to their specific physiological characteristics, such as few sweat glands, thick coats of fur, high metabolic rate, and growth rate [3,4,5].

The seasonal effects on growth performances, reproduction traits, and meat quality of rabbits have been extensively reported in previous studies. For instance, Ayyat et al. noted that the body weight, daily weight gain, and feed consumption of rabbits reared during the summer were significantly lower than those reared during the winter [6]. Attia et al. indicated that rabbits fed in autumn and winter had a greater growth rate and a better feed conversion ratio than those fed in summer [7]. Seasonal changes cause alterations in ambient conditions such as air temperature and relative humidity that have potential impacts on the nutrients metabolism and growth hormone secretion that consequently result in growth performance variations [8,9]. In general, seasonal alterations influence the reproduction traits of both female and male rabbits. Regarding buck rabbits, in spring the values of sperm concentration and production were significantly higher than in autumn, while in autumn the percentage of motile and progressive cells was significantly higher than in summer [10,11]. As for doe rabbits, a greater ovulation rate was seen in both winter and spring than in both summer and autumn [12]. Moreover, a lower number of pups born and born alive per delivery were observed in both summer and autumn compared to both winter and spring [13]. These seasonal variations in reproductive performances might be explained by the modulatory roles of photoperiodicity in the hypothalamus–pituitary axis and related hormone release [14,15]. The meat quality of rabbits is the most important trait that directly affects consumer acceptance and purchase willingness, which also exhibit seasonal differences. María et al. demonstrated that seasons exerted significant effects on meat color, pH, water holding capacity, and meat texture parameters and pointed out that the altered energy demands among different seasons had impacts on normal metabolic processes in skeletal muscles, which would further affect meat quality traits [16].

Gut microbiota have profound influences on the production performance and health status of farm animals, owing to their crucial roles in maintaining metabolic homeostasis as well as modulating physiological, neurological, and immunological functions [17]. Previously, several studies have investigated relationships among the gut microbiota, productivity, and well-being of rabbits. Combes et al. indicated that gut microbiota transferred from doe rabbits to their offspring could promote the development of the immune system of pups and effectively influence their survival rate [18]. Fang et al. found that the butyrate-producing bacteria, which belonged to the family Ruminococcaceae, were positively associated with both weaning weight and finishing weight, which was linked to the important regulation roles of these bacteria in energy metabolism and immune response [19,20]. Liu et al. elucidated that the significant correlations between Ruminococcaceae members and amino acid metabolism in *longissimus dorsi* muscle could affect meat quality [21]. In addition, El-Aziz et al. suggested that the increased abundance of *Lactobacillus* was beneficial to improving hematobiochemical parameters and reducing pathogenic bacteria growth [22], while Xu et al. revealed that Eimeria intestinalis infection destroyed the intestinal homeostasis at the parasitized sites, resulting in *Akkermansia* reduction and lipid metabolism disorder, which would exacerbate intestinal inflammation [23].

To our best knowledge, the seasonal variations in production performance, health status, and gut microbiota of meat rabbits reared in a half-sealed housing system were first determined in the present study. Our findings would give insight into optimizing management and feeding strategies to improve the productivity and well-being of meat rabbits in different seasons and provide basic knowledge for mechanistic studies of adaptation physiology.

## 2. Materials and Methods

### 2.1. Experimental Animals and Management Practices

This study was conducted from November to January (winter) and June to August (summer) in Fuzhou, Fujian Province, at a commercial meat rabbit farm during the years 2022–2023. In each season, seventeen Ira rabbits (9 males and 8 females) at 45 ± 3 days old with a similar body weight (1.2 ± 0.1 kg) were randomly selected and reared in the same semi-confined rabbit house, which was both mechanical and naturally ventilated. In each season, during the 45-day experimental period, the ambient temperature and relative humidity were recorded daily at 8:00 a.m. and 4:00 p.m.; the mean values are shown in Appendix A. All rabbits were exposed to natural lighting, and the photoperiod was determined by sunrise and sunset times throughout the experimental period. All rabbits accessed water ad libitum and were fed the same commercial pelleted feed (Appendix A) throughout the experimental period, but batch fed at 5:30 a.m. and 5:30 p.m. in summer and batch fed at 6:30 a.m., 12:30 a.m., and 6:30 p.m. in winter.

### 2.2. Sample Collection and Phenotype Measurement

At the end of the experiment, blood samples were collected from the ear vein of each rabbit into tubes containing EDTA and subjected to centrifugation at 3000 rpm for 20 min at room temperature to obtain plasma. The separated plasma was stored at −80 °C until subsequent assays. The slaughter weight (SW) of experimental rabbits was measured before electro-anesthesia and sacrifice by jugulation. After full bleeding, the pelt, viscera, and tail were removed, and the carcass was weighed to obtain carcass weight (CW). The hard fecal samples were collected from the rectum and stored at −80 °C until further processing.

### 2.3. Health Status Indictors Measurement

The plasma concentrations of heat shock protein 90 (HSP90) and Interleukin-2 (IL-2) were determined by a commercial ELISA kit purchased from Jiangsu Meimian Industry Co., Ltd. (Yancheng, China). The activities of superoxide dismutase (SOD) and glutathione (GSH) and levels of malondialdehyde (MDA), Interleukin-1α (IL-1α), and Interleukin-1β (IL-1β) were analyzed by commercial ELISA kits purchased from Nanjing Jiancheng Bioengineering Institute (Nanjing, China).

### 2.4. 16 S rRNA Gene Sequencing Analysis

Total microbial genomic DNA was extracted from hard fecal samples using the QIAamp Fast DNA Stool Mini Kit (QIAGEN, Hilden, Germany) according to the manufacturer’s instructions. The quantity and quality of DNA were detected by using the Nanodrop ND-2000 spectrophotometer (Thermo Fisher Scientific, Waltham, MA, USA) and 1.5% agarose gel electrophoresis, respectively. The V3–V4 hypervariable regions of the 16S rRNA gene were amplified by the barcoded fusion primers 341F (5′-CCTACGGGNGGCWGCAG-3′) and 806R (5′-GGACTACHVGGGTATCTAAT-3′) and subjected to sequencing on the Hiseq-2500 platform (Illumina, San Diego, CA, USA).

QIIME (v.1.9.1) were used for the quality control process of sequencing data, including filtering out primers, barcodes, and low-quality sequences (quality score < 20). High-quality paired-end reads were merged into tags by using FLASH (v.1.2.11). In order to normalize the sequencing depth, the library size of microbial sequences was restricted to 50,000 tags per sample. USEARCH (v.10.0) was used to cluster tags into operational taxonomic units (OTUs) at 97% sequence similarity. OTU taxonomic category assignments were performed using the SILVA database (v.132). The alpha and beta diversity indices were calculated using Mothur (v.1.41.1) and QIIME (v.1.9.1), respectively. The potential functional profiles of microbial communities were predicted using phylogenetic investigation of communities by reconstruction of unobserved states v.2.0 (PICRUSt2).

### 2.5. Statistical Analysis

Student’s *t*-test with a false discovery rate (FDR) correction was used to analyze seasonal variations in growth performances and health status indicators. Redundancy analysis (RDA) was performed to identify the effect of seasonal changes on the gut microbiota. Principal coordinate analysis (PCoA) was performed using both unweighted and weighted UniFrac distances to show seasonal alterations in gut microbial structures. A Wilcoxon test with FDR correction was applied to determine seasonal differences in alpha diversity, beta diversity, and relative abundances of different microbial taxa. The seasonal changes in functional capacities were characterized by a logistic regression analysis. The seasonal differentially enriched OTUs were identified by Linear discriminant analysis of Effect Size (LEfSe). Mantel test analysis was conducted to assess the correlations among production performance, health status indicators, and gut microbiota.

## 3. Results

### 3.1. Seasonal Differences in Production Performance and Health Status of Rabbits

As shown in Table 1, slaughter weight (SW) and carcass weight (CW) observed in winter were significantly higher than those obtained in summer (*p* < 0.01). Regarding potential heat stress, the level of HSP90 was significantly increased in the summer compared to the winter (*p* < 0.01). As for possible oxidative stress, the greater the MDA concentration, the declined GSH and SOD activities measured in the summer compared with those assayed in the winter (*p* < 0.01). Moreover, significant elevations of inflammation status-related biomarkers IL-1α, IL-1β, and IL-2 were found in the summer (*p* < 0.01).

### 3.2. Seasonal Changes in Gut Microbial Structure and Composition of Rabbits

To assess the effect of season on gut microbial alternations in rabbits, RDA analysis was first performed. The result showed that season played more important roles in gut microbial structure variation than gender and age (Appendix A). The alpha diversity of the gut microbiota differed remarkably between summer and winter (Table 2). Both the Shannon and ACE indexes of the gut microbiota in the summer were significantly lower than those in the winter (*p* < 0.05). In terms of beta diversity, PCoA analysis based on Unweighted and Weighted Unifrac distance was performed to determine the differences between summer and winter. The result indicated that samples from different seasons were clustered separately (Figure 1A,B). Additionally, gut microbial Unweighted and Weighted Unifrac distances in the summer were significantly higher than those in the winter (Figure 1C, *p* < 0.05).

Gut microbial community composition analysis showed that the dominant phyla of gut microbiota in all samples were Firmicutes (53.66%), Bacteroidetes (27.05%), Verrucomicrobiota (10.11%), Proteobacteria (1.56%), Actinobacteriota (1.24%), and Desulfobacterota (1.07%) (Figure 2A). Despite this, the relative abundances of these dominant phyla changed obviously in different seasons. The relative abundance of Firmicutes was significantly increased from summer to winter (*p* < 0.05). The Verrucomicrobiota in winter accounted for significantly higher proportions than that in summer (*p* < 0.05). The alterations in the relative abundances of Proteobacteria, Actinobacteriota, and Desulfobacterota were consistent and possessed significantly higher values in summer than in winter (*p* < 0.05). At the genus level, *Akkermansia*, *Oscillospiraceae NK4A214 group*, *Christensenellaceae R-7 group*, *Ruminococcus*, *Alistipes*, *Oscillospiraceae V9D2013 group*, *Monoglobus*, *Bacteroides*, and *Eubacterium siraeum group* were the top nine genera. Among these, *Christensenellaceae R-7 group* and *Monoglobus* were significantly enriched in summer, while *Akkermansia*, *Oscillospiraceae NK4A214 group*, and *Alistipes* were more abundant in winter (Figure 2B, *p* < 0.05).

To explore more microbial taxa abundance variations in different seasons, the OTU data were analyzed. As seen from Figure 3, 40 OTUs were significantly affected by season, including the abundance of 23 OTUs in the summer and the remaining OTUs in the winter. Amongst summer-enriched OTUs, 7 OTUs belonged to Muribaculaceae, 5 OTUs were annotated to Lachnospiraceae, 2 OTUs belonged to *Monoglobus*, and 1 OTU belonged to each of the *Christensenellaceae R-7 group*, *Ruminococcus sp.NK3A76*, *Phascolarctobacterium*, *Tyzzerella*, *Eubacterium siraeum group*, *Ruminococcaceae Incertae Sedis*, Oscillospirales UCG-010, and Atopobiaceae. Amongst winter-enriched OTUs, 2 OTUs were classified as each of *Akkermansia*, *Oscillospiraceae NK4A214 group*, Clostridia UCG-014, and Clostridia vadinBB60 group, and 1 OTU was classified as each of *Akkermansia muciniphila*, *Bacteroides sp.Marseille-P3166*, *Bacteroides caccae*, *Eubacterium coprostanoligenes group*, *Lachnospiraceae NK4A136 group*, Barnesiellaceae, Lachnospiraceae, Oscillospirales UCG-010, and Eubacteriaceae.

### 3.3. Seasonal Alterations in Gut Microbial Functions of Rabbits

To investigate the functional differences in the gut microbiota of rabbits in different seasons, functional prediction analysis was performed using PICRUSt2, and then the predicted functional categories were subjected to logistic regression analysis. The result showed that four significantly different Kyoto Encyclopedia of Genes and Genomes (KEGG) pathways are present in summer, including the biosynthesis of other secondary metabolites, the metabolism of terpenoids and polyketides, infectious diseases, and xenobiotic biodegradation and metabolism (Figure 4). There were another four significantly different KEGG pathways exhibited in winter, namely replication and repair, amino acid metabolism, glycan biosynthesis and metabolism, and cell motility.

### 3.4. Associations of Gut Microbial Variations with Host Phenotypic Alterations in Different Seasons

Mantel test analysis was conducted to evaluate the correlations between gut microbial variations and host phenotypic alterations in different seasons. As shown in Figure 5, summer-enriched microbes showed strong negative correlations with both SW and CW, while winter-enriched microbes had strong positive correlations with the two traits. As for health status-related biomarkers, both GSH and SOD were positively associated with winter-enriched microbes but were negatively associated with summer-enriched microbes. In contrast, IL-1β was positively correlated with summer-enriched microbes but was negatively correlated with winter-enriched microbes. Meanwhile, summer-enriched microbes had a specific positive association with HSP90, and winter-enriched microbes had a specific negative association with IL-1α. In addition, summer-enriched functions were found to have negative relations with both GSH and SOD, while positive relations between winter-enriched functions and the two biomarkers were observed. On the other hand, close associations between production performance and health status indicators were observed. For instance, CW was positively associated with SW, and they had consistent relationships with all health status-related biomarkers. GSH possessed a positive correlation with SOD, and both of them were negatively linked to IL-1α and IL-1β.

## 4. Discussion

Ambient conditions, management practices, photoperiod, and disease-related factors determine the degree of seasonal effects on livestock’s production performance on commercial farms. Thus, a holistic understanding of the seasonal patterns of farm animals’ physiology and production performance helps to formulate management and feeding manipulation strategies that will improve productivity and profitability. In this study, we investigated the seasonal alterations in production performance, health status, and gut microbial communities of meat rabbits reared in a semi-confined barn.

Slaughter weight and carcass weight are important economic traits in the meat rabbit industry, which faces seasonal fluctuations. Previous studies revealed that the summer season with high ambient temperature and relative humidity adversely affected the maintenance of thermal balance and resulted in physiological homeostasis alterations, which decreased the slaughter weight and carcass weight of rabbits [6,24]. Indeed, a significantly declining slaughter weight and carcass weight were observed in the summer in the present study. Moreover, disturbances in physiological status, including thermal tolerance characteristics, antioxidant properties, and immune responsiveness, were seen in meat rabbits reared in the summer. Rabbits possess fewer sweat glands and thicker fur, leading to the heat dissipation issue, which makes them more sensitive to high ambient temperature [3]. HSP90 is a highly abundant and ubiquitous molecular chaperone that exerts regulatory roles in diverse biological processes and has been reported to be associated with heat tolerance traits in farm animals [25,26]. In this study, the average daily temperature of the semi-confined housing system in summer was 28.5 °C, exceeding the thermoneutrality range (18–24 °C) for rabbits, which may cause an elevation of the HSP90 level to prevent potential heat stress injury. Under thermoneutral conditions, antioxidative enzymes such as SOD and GSH are capable of scavenging reactive oxygen species to modulate redox status and dynamic balance. In agreement with our findings, both Saghir et al. and Madkour et al. pointed out that the rabbits’ plasma concentrations of SOD and GSH declined with the increasing house temperature while the level of MDA was elevated, which was linked to progressive oxidative stress [27,28]. Simultaneously, considering the intimate relationship between oxidative stress and inflammation responses, we noted that the levels of pro-inflammatory cytokines such as IL-1α, IL-1β, and IL-2 were increased. In addition, the humid circumstances of the summer season are regarded as one of the most significant risk factors that trigger the dermatophytoses and respiratory diseases of rabbits, which may further aggravate the burden on the immune system [29,30].

The gut microbiota exerts modulatory roles in metabolic pathways, immune processes, and neural functions, which have great influences on the health and production performance of farm animals [31]. Meanwhile, different factors such as genetic background, gender differences, age variations, nutritional levels, and management practices effectively affected gut microbial diversity, composition, and functions [32,33]. In this study, RDA analysis could mirror seasonal effects on gut microbiota alterations, owing to the fact that meat rabbits with similar ages and sex ratios fed the same feed pellets were investigated. Hence, we speculated that differences in ambient control and management regimes in different seasons should make major contributions to gut microbial seasonal variations. For example, compared to the winter season, the rabbits raised in the summer season had significantly lower and higher alpha and beta diversity, respectively. This may be explained by the relative high ambient temperature and humidity exposure in summer, which significantly reduced feed intake and further resulted in a shift in gut microbial diversity [34,35]. Changes in gut microbiota at different taxonomic levels between summer and winter were also observed. In winter, the relative abundances of Firmicutes and Verrucomicrobiota were significantly increased, whereas Proteobacteria, Actinobacteriota, and Desulfobacterota were markedly decreased. Gut microbiota is regarded as a key factor orchestrating energy homeostasis during increased energy demand under relatively low air temperature conditions [36]. Accordingly, more abundant phylum Firmicutes and Verrucomicrobiota linked to energy harvest should be an adaptive response to energy scarcity in winter [37,38]. Moreover, the thriving growth of these two phyla might inhibit the proliferation of Proteobacteria, Actinobacteriota, and Desulfobacterota via nutritional and geographical competitive behaviors [39]. To further detect the differences in relative abundances of different microbial taxa, the genus and OTU data were analyzed. As expected, both the genus *Akkermansia* and the species *Akkermansia muciniphila*, which belonged to the phylum Verrucomicrobiota, were enriched in the winter season. *Akkermansia muciniphila* is a well-known mucin-degrading bacterium capable of converting mucins to acetate [40]. The acetate can be metabolized in peripheral tissues to meet energy demands or converted to ketone bodies in gut epithelial cells or hepatocytes, serving as an alternative fuel to glucose [41,42]. However, we also found some discrepancies in microbial enrichment between phylums and lower taxonomic levels. For instance, both the genus *Oscillospiraceae NK4A214 group* and the *Christensenellaceae R-7 group* derived from Firmicutes, which were augmented in winter and summer, respectively. The lighting regime between summer and winter was different, and previous studies reported that illumination time variation significantly affected the relative abundances of these two genera [43,44]. Additionally, *Alistipes* and Muribaculaceae were members of Bacteroidetes; the former showed greater abundance in winter and the latter exhibited higher abundance in summer. The differences in feeding regime between summer and winter might lead to the differential prevalence of these two genera [45,46]. In agreement with the results of microbial diversity and composition analysis, functional pathways linked to microbial proliferation (replication and repair, cell motility) and nutrient metabolism (amino acid metabolism, glycan biosynthesis, and metabolism) were enriched in the winter season, while those associated with environmental adaptive responses (biosynthesis of other secondary metabolites, metabolism of terpenoids and polyketides, infectious diseases, xenobiotic biodegradation, and metabolism) were more abundant in the summer season.

To investigate the relationships between gut microbiota and host productivity and health status, a mantel test analysis was performed. On the one hand, strong inter-system interactions were observed. For instance, the winter-enriched microbes showed positive associations with slaughter weight and carcass weight, while they exhibited negative associations with IL-1α and IL-1β. Earlier studies could provide some evidence for these results. Firmicutes have vital roles in dietary protein and carbohydrate metabolism; both Liu et al. and Li et al. demonstrated that the increased abundance of Firmicutes positively affected the growth performances of meat rabbits [21,47]. *Akkermansia muciniphila* is a promising candidate probiotic that has beneficial effects on host immunologic and metabolic functions [48]. The causal evidence of its anti-inflammatory property via reducing the levels of inflammatory cytokines has been reported in a variety of mouse model studies [49,50]. On the other hand, the close intra-system interactions indicated that our findings were reliable, such as that slaughter weight was intimately associated with carcass weight and the different stress status indicators were positively or negatively correlated with production performance.

Based on our findings, several realistic strategies can be applied to ameliorate the unfavorable effects of seasonal alterations on productivity, health status, and the gut microbial community of meat rabbits. Firstly, considering the relative high ambient temperature and humidity in summer could cause disturbance of gut microbiome, potential heat stress, reduced antioxidant defense, and increased inflammation risk, probiotics (e.g., *Bacillus subtilis* and *Clostridium butyricum*) and prebiotics (e.g., fructooligosaccharides and mannan-oligosaccharides) interventions should be an effective solution for solving these concerns due to their beneficial roles in maintaining gut microbial homeostasis, alleviating heat stress, improving antioxidative status, and modulating immune responses. Moreover, meat rabbits raised in winter mainly face the energy demand challenge, and optimizing feed formulation and feeding pattern should be regarded as available approaches. Last but not least, utilizing smart systems such as sensor technologies in meat rabbit production can monitor the physiological parameters or adaptation physiology, which is helpful in improving both profitability and animal welfare.

The observations of the present study are meaningful but limited by the small sample size. Thus, it is essential to further examine the impacts of seasonal changes on production performance, health status, and gut microbiota in a larger meat rabbit population. Furthermore, future studies using multi-omics techniques such as epigenome, transcriptome, proteome, and metabolome in conjunction with a variety of productivity and physiological parameters obtained from four seasons will provide systematic and comprehensive insights into molecular mechanisms that allow meat rabbits to maintain physiological homeostasis in different seasons and lay solid foundations for advancing management practices.

## 5. Conclusions

In conclusion, remarkably different changes in production performance, health status, and gut microbiota of meat rabbits reared in semi-confined conditions were seen between summer and winter. These observations should be closely linked to meat rabbits adapted to variations in ambient conditions, energy demand, photoperiod, and feeding regime in different seasons. Importantly, our study not only proposed several available strategies to alleviate the negative effects of seasonal alterations on the productivity and well-being of meat rabbits but also pointed out the future directions for this study of molecular mechanisms in adaptation physiology.

## Figures and Tables

**Figure 1 animals-14-00113-f001:**
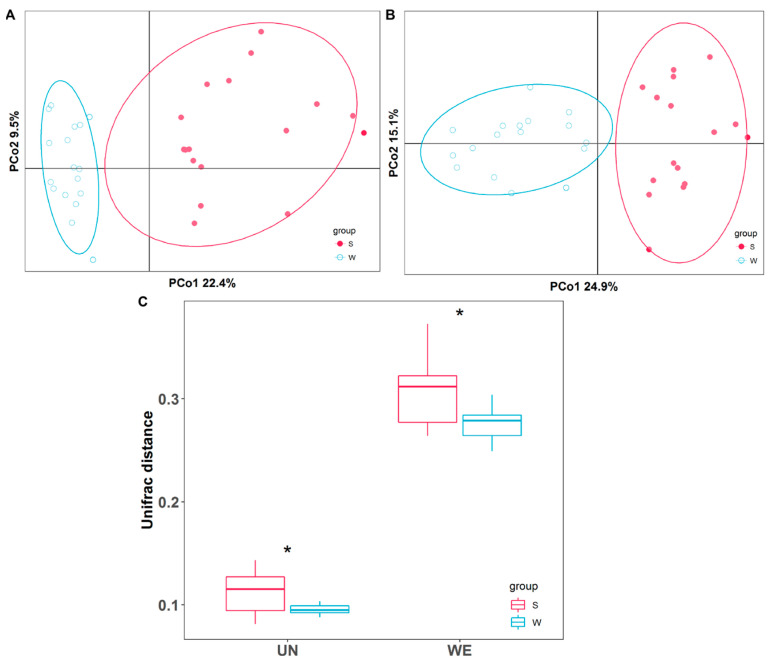
Gut microbial beta diversity comparison analysis between summer and winter. (**A**) PCoA analysis based on Unweighted Unifrac distance. (**B**) PCoA analysis based on Weighted Unifrac distance. (**C**) Differences in Unweighted (UN) and Weighted (WE) Unifrac distance metric between summer and winter. S: summer, W: winter “*” represents *p* < 0.05.

**Figure 2 animals-14-00113-f002:**
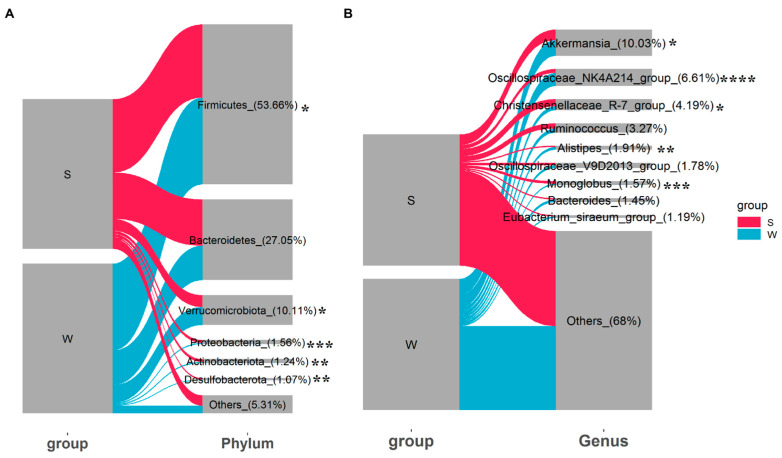
The variations in relative abundances of gut microbiota at (**A**) phylum level and (**B**) genus level between summer and winter. “*” represents *p* < 0.05, “**” represents *p* < 0.01, “***” represents *p* < 0.005, and “****” represents *p* < 0.001.

**Figure 3 animals-14-00113-f003:**
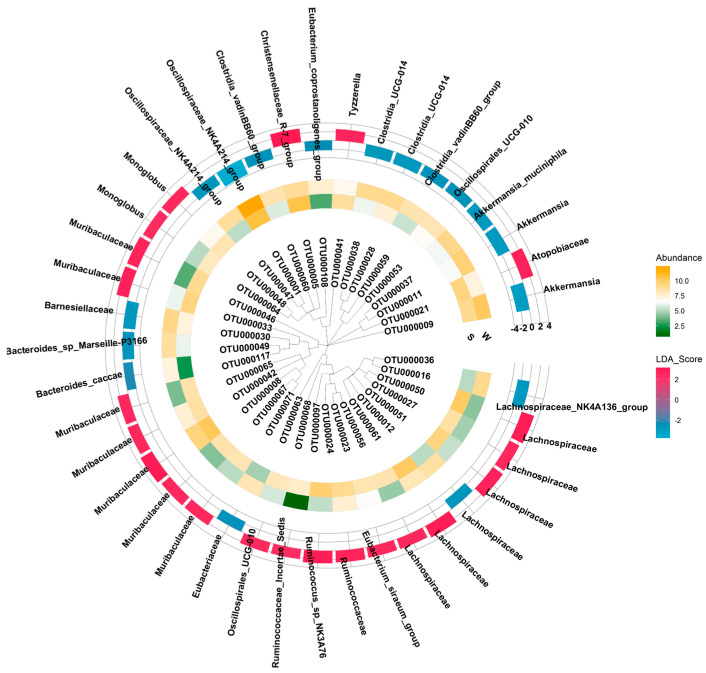
Seasonal differentially enriched OTUs The plot is composed of three parts as follows: the phylogenetic tree of OTUs, the heatmap, which denotes the abundances of OTUs, and the barplot, which represents the LDA score of OTUs. The positive LDA value means OTUs are enriched in summer, and the negative LDA value means OTUs are enriched in winter.

**Figure 4 animals-14-00113-f004:**
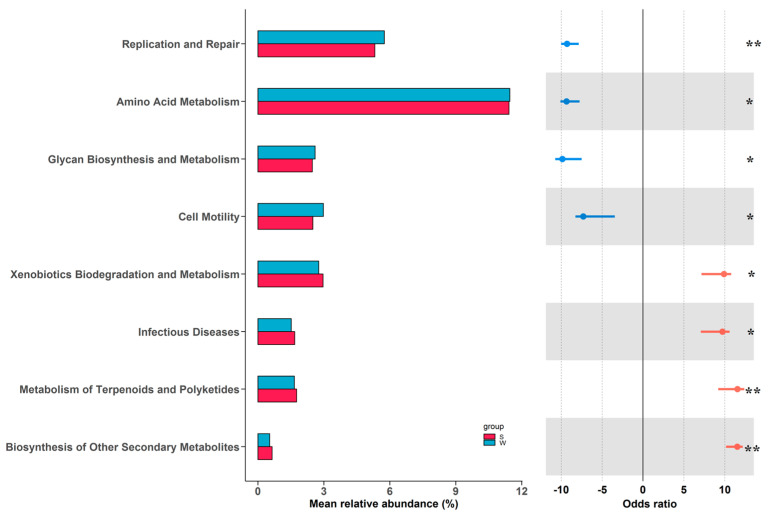
Seasonal alterations in gut microbial functional capacities. “*” represents *p* < 0.05, and “**” represents *p* < 0.01.

**Figure 5 animals-14-00113-f005:**
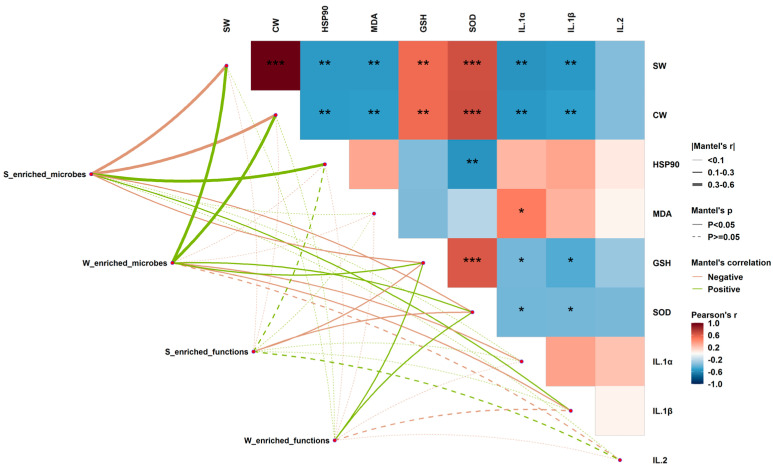
Correlations among productivity, health status, and gut microbial indicators. Edge width and color correspond to Mantel’s r statistic for the corresponding distance correlations among gut microbiota, production performance, and health status, and edge type denotes the statistical significance. The heatmap denotes Pearson’s correlation coefficients between productivity and health status indicators. “*” represents *p* < 0.05, “**” represents *p* < 0.01, and “***” represents *p* < 0.005.

**Table 1 animals-14-00113-t001:** Production performance and health status indicators of meat rabbits in the summer and winter.

Indictors	Summer	Winter	*p* Value
SW (g)	1879.41 ± 109.06	2650 ± 272.71	7.76 × 10^−7^
CW (g)	1408.82 ± 79.52	1941.18 ± 203.28	1.09 × 10^−6^
HSP90 (ng/L)	325.62 ± 88.43	233.22 ± 52.39	9.88 × 10^−4^
MDA (nmol/mL)	5.23 ± 1.5	3.72 ± 1.13	7.6 × 10^−3^
GSH (U/mL)	245.80 ± 129.99	435.04 ± 130.39	2.27 × 10^−3^
SOD (U/mL)	620.99 ± 59.54	783.88 ± 105.43	4.15 × 10^−5^
IL-1α (ng/L)	6.63 ± 1.49	4.75 ± 1.39	2.3 × 10^−3^
IL-1β (ng/L)	38.26 ± 9.24	26.88 ± 11.07	2.58 × 10^−3^
IL-2 (ng/L)	616.86 ± 251.42	361.87 ± 198.54	5.54 × 10^−3^

Note: The value present in the table was mean ± sd. SW: slaughter weight, CW: carcass weight, HSP90: heat shock protein 90, MDA: malondialdehyde, SOD: superoxide dismutase, GSH: glutathione, IL-1α: Interleukin-1α, IL-1β: Interleukin-1β, and IL-2: Interleukin-2.

**Table 2 animals-14-00113-t002:** Gut microbial alpha diversity of meat rabbits in summer and winter.

Indictors	Summer	Winter	*p* Value
Shannon	6.88 ± 0.42	7.25 ± 0.35	1.77 × 10^−2^
ACE	1676.76 ± 405.74	1874 ± 154.81	1.74 × 10^−3^
Good’s Coverage	0.994 ± 0.002	0.994 ± 0.001	3.39 × 10^−1^

Note: The value present in the table was mean ± sd.

## Data Availability

The data presented in this study are available on request from the corresponding author. The data are not publicly available due to [the data reflect the specific production performance of the enterprise that is covered by the privacy policy].

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
