# Peer review of "Seasonal Variations in Production Performance, Health Status, and Gut Microbiota of Meat Rabbit Reared in Semi-Confined Conditions"

_animals, 2023, doi:10.3390/ani14010113_

Round 1
Reviewer 1 Report
Comments and Suggestions for Authors
In this article, the authors investigated the differences in production performance, health status, and gut microbiota of meat rabbits between summer and winter. As in previous studies, the production performance of meat rabbits improved more in winter than in summer. Furthermore, the dominant phyla of the gut microbiota were the same throughout the year, but the relative abundance of these dominant phyla clearly changed in different seasons. The results presented in this study provide important information for considering strategies to be taken in making environmental improvements in meat rabbit fattening.
This study presents useful information for breeding rabbits for meat, but I think this manuscript needs to be considered more sufficiently before publication.
1) Line 368-372. To ensure the novelty and significance of this study, it is desirable to present experimental data showing productivity improvement in meat rabbits by intervention of probiotics and prebiotics.
2) Result. Please review the item number in the results section. "3.3." is missing.
3) Line 287. Please clarify what the numbers “34518931” indicate.
Author Response
1.Line 368-372. To ensure the novelty and significance of this study, it is desirable to present experimental data showing productivity improvement in meat rabbits by intervention of probiotics and prebiotics.
Response: Thank you for your comment. Due to the probiotics and prebiotics intervention experiment have not been performed in the present study, we are unable to provide these experimental data.
2.Result. Please review the item number in the results section. "3.3." is missing.
Response: Thank you for your comment. We have defined the wrong item numbers and they are modified in the revised manuscript.
3.Line 287. Please clarify what the numbers “34518931” indicate.
Response: Thank you for your comment. It is the PMID number of “reference [3]” and we have deleted it in the revised manuscript.
Reviewer 2 Report
Comments and Suggestions for Authors
The presented publication does not introduce significant scientific innovations; however, it holds particular importance for breeders engaged in the seasonal growing of rabbits. It is evident that environmental changes directly impact rabbit indicators, a fact informatively and comprehensively presented through tables and graphs by the authors. Nevertheless, the article appears to be written in an informal manner, lacking scientific language, specificity, or an understanding of how certain sections should be presented.
A few specific observations:
- The abstract resembles more of a summary of results, lacking essential elements. The frequent use of "here" in the first line and throughout the text is inappropriate. It needs specification, for instance, "in a recent study," "in this study," or similar expressions.
- The term "meat" is repetitively used in sentences, such as in lines 59–60, "meat quality of meat rabbits." In my opinion, constant repetition of "meat rabbits" is unnecessary. The same comment applies to lines 76–77, "Meat quality of meat rabbits."
- In lines 100–105, firstly, the style and, secondly, the aim of the study are not presented in a manner consistent with fundamental scientific article writing rules.
- The "Materials and Methods" section is written clearly, but I believe that providing a detailed composition of commercial feed is crucial.
- Lines 171–172 should have abbreviation explanations. I believe explanations should be indicated in the notes in Table 1 as well.
- In the discussion, there is no need to reference tables; this has already been done in the results section.
- The conclusions need a complete rewrite, as they currently lack substance. It is essential to specifically state the key and significant results obtained during the authors' experiment.
In conclusion, I do not believe that the current version of the article meets the standards for the Animals journal. I recommend allowing resubmission after the article has been rewritten and improved.
Comments on the Quality of English LanguageCould be improved.
Author Response
1.The abstract resembles more of a summary of results, lacking essential elements.
Response: Thank you for your comment. Because of the Animals journal has the words limitation in the abstract, we have refined our expression to provide enough information.
2.The frequent use of "here" in the first line and throughout the text is inappropriate. It needs specification, for instance, "in a recent study," "in this study," or similar expressions.
Response: According to this comment, we have modified the expression pattern.
3.The term "meat" is repetitively used in sentences, such as in lines 59–60, "meat quality of meat rabbits." In my opinion, constant repetition of "meat rabbits" is unnecessary. The same comment applies to lines 76–77, "Meat quality of meat rabbits."
Response: According to this comment, we have modified these descriptions.
4.In lines 100–105, firstly, the style and, secondly, the aim of the study are not presented in a manner consistent with fundamental scientific article writing rules.
Response: Thank you for your comment. Considering different authors may have different writing styles and expression pattern, we thought the aim of the present study has been appropriately described.
5.The "Materials and Methods" section is written clearly, but I believe that providing a detailed composition of commercial feed is crucial.
Response: Thank you for your comment. We have provided the detailed composition of commercial feed in Table S2.
6.Lines 171–172 should have abbreviation explanations. I believe explanations should be indicated in the notes in Table 1 as well.
Response: According to this comment, we have added the explanations for these abbreviations in the note of Table 1.
7.In the discussion, there is no need to reference tables; this has already been done in the results section.
Response: According to this comment, we have deleted these references to tables and figures.
The conclusions need a complete rewrite, as they currently lack substance. It is essential to specifically state the key and significant results obtained during the authors' experiment.
Response: Thank you for your comment. The sentence “the remarkably changes in production performance, health status, and gut microbiota of meat rabbit reared in semi-confined conditions were seen between summer and winter” highly concluded the results of the present study. Moreover, specifically state the key and significant results will make the conclusion more similar to the abstract.
Reviewer 3 Report
Comments and Suggestions for Authors
Paper review:
Seasonal variations in production performance, health status, and gut microbiota of meat rabbit reared in semi-confined conditions.
Line 31: it would be beneficial to indicate that HSP90 is a protein. Do the same with pro inflammatory cytokines.
Line 33: Alpha and Beta diversity of what? I recommend to indicate that you are talking about different microbial taxa.
Line 61, 63, 78, 89, 92, 95, 97, 295, 355: when citing authors, it is essential to include the year of publication for each reference.
Line 125: I recommend using the abbreviation “(SW)”.
Line 171: what does MDA mean?
Line 172: what do the abbreviations GSH and SOD stand for?
Line 175: in the notes of table 1, I recommend include the meaning of HSP90, MDA, GSH, SOD.
Line 235: what does PICRUSt2 mean?
Line 237: What do you mean by KEGG?
Materials and methods
Detailed information about the management practices, feeding regimen, and housing conditions is provided, which is crucial for replication the study.
The section is heavy with technical terms.
Results
The section presents comprehensive data on various parameters (supported by tables and figures for better understanding).
Overall, this section is thorough and well-presented, some readers might benefit from more direct explanations or summaries of what these results mean in a broader context.
Conclusion
The section effectively summarizes the significant changes observed in production performance, health status, and gut microbiota of meat rabbits across seasons.
While the section proposes strategies for addressing seasonal effects, it could be more specific in detailing these strategies.
Author Response
1.Line 31: it would be beneficial to indicate that HSP90 is a protein. Do the same with pro inflammatory cytokines.
Response: According to this comment, we have modified the sentence.
2.Line 33: Alpha and Beta diversity of what? I recommend to indicate that you are talking about different microbial taxa.
Response: Thank you for your comment. Shannon and ACE index for Alpha diversity. Unweighted and Weighted Unifrac distance for Beta diversity. However, we can not describe these indices in detail, due to the words limitation of the abstract in the Animals journal.
3.Line 61, 63, 78, 89, 92, 95, 97, 295, 355: when citing authors, it is essential to include the year of publication for each reference.
Response: Thank you for your comment. We have cited the references according to the style requirement of the Animals journal.
4.Line 125: I recommend using the abbreviation “(SW)”.
Response: According to this comment, we have modified the sentence.
5.Line 171: what does MDA mean?
Response: Thank you for your comment. MDA represents for malondialdehyde and we have given the explanations for common indictors in the “Materials and Methods” section.
6.Line 172: what do the abbreviations GSH and SOD stand for?
Response: Thank you for your comment. We have given the explanations for common indictors in the “Materials and Methods” section.
7.Line 175: in the notes of table 1, I recommend include the meaning of HSP90, MDA, GSH, SOD.
Response: According to this comment, we have added the explanations for these abbreviations.
8.Line 235: what does PICRUSt2 mean?
Response: Thank you for your comment. PICRUSt2 represents for phylogenetic investigation of communities by reconstruction of unobserved states v.2.0 and we have provided the information in the“Materials and Methods” section of the revised manuscript.
9.Line 237: What do you mean by KEGG?
Response: Thank you for your comment. KEGG represents for Kyoto Encyclopedia of Genes and Genomes and we have provided the information in Line 237.
10.While the section proposes strategies for addressing seasonal effects, it could be more specific in detailing these strategies.
Response: Thank you for your comment. These strategies have been described in detail in Line 370-383.
Round 2
Reviewer 2 Report
Comments and Suggestions for Authors
The article was improved according to my comments.